# Social, Psychological, and Philosophical Reflections on Pandemics and Beyond

**Abraham Rudnick**

Department of Psychiatry and School of Occupational Therapy, Dalhousie University,
Halifax, NS B3H 4R2, Canada; harudnick@hotmail.com

**Abstract:** This conceptual paper presents social, psychological and philosophical (ethical and epistemological) reflections regarding the current (COVID-19) pandemic and beyond, using an analytic and comparative approach. For example, Taiwan and Canada are compared, addressing Taiwan's learning from SARS. Suggestions are made in relation to current and future relevant practice, policy, research and education. For example, highly exposed individuals and particularly vulnerable populations, such as health care providers and socially disadvantaged (homeless and other) people, respectively, are addressed as requiring special attention. In conclusion, more reflection on and study of social and psychological challenges as well as underlying philosophical issues related to the current pandemic and more generally to global crises is needed.

**Keywords:** education; pandemic; philosophy; policy; practice; psychology; research; social

---

## 1. Introduction

Societies are measured in part in relation to how they rise to the occasion of collective crises and learn from them. For example, both Taiwan and Canada (specifically Toronto) were similarly directly impacted by the Severe Acute Respiratory Syndrome (SARS) pandemic and related nosocomial (hospital-based) viral transmission a couple of decades ago [1], yet it seems that Taiwan learned from that to prepare well for such pandemics, whereas Canada (including Toronto) did not [2]. The current Coronavirus Disease 2019 (COVID-19) pandemic is such a crisis and raises various problems that are insufficiently addressed to date (such as the impact of international travel on global health), some of them reflective of underlying social and other challenges across the world [3]. In addition to medical and technological problems, social as well as psychological problems and underlying philosophical (particularly ethical and epistemological) challenges have to be better addressed to further improve the approach to this pandemic and arguably to future pandemics and other global crises. In this conceptual paper, I use an analytic [4] and comparative [5] approach to present related social, psychological, and philosophical issues, using my experience and expertise as a social scientist and health researcher [6], a clinically practicing psychiatrist, a health care administrator [7,8], and a philosopher of health and related care [9,10]. I conclude with practice and policy as well as research and education suggestions.

## 2. Social and Psychological Reflections

The current pandemic poses important social challenges. For example, many people have been laid off work temporarily or permanently during the pandemic due to an insufficient workload, such as in the service sector. Unemployment is associated with disrupted mental well-being [11] and with other personal as well as societal disruptions such as poverty, crime, and more. The most vulnerable to such disruptions are typically people who are already disadvantaged, such as those from lower socioeconomic strata and many retired people. Hence, the general population, and particularly vulnerable populations such as socially disadvantaged people (homeless individuals and others),

may require particular attention during and soon after the pandemic to try to ensure that they are at least not further disadvantaged. Another example is the expected political disruption during a pandemic, particularly in countries where the regime is not democratically robust (such as in Israel where the prime minister is allowed to stay in office in spite of incurring criminal charges [12]). In such countries, some people may use the opportunity during the pandemic to disorganize society or to further restrict the general public and/or special social groupings that are considered by them as socially undesirable (such as racialized minorities and others). Such disorganization and restriction can further disrupt personal and societal well-being during the pandemic (and after it if the disruptive political changes remain in place). Hence, the general public and/or special social groupings that are considered socially undesirable by some may require particular attention during and after the pandemic to try to support them in relation to pandemic-related disruptive political change.

The current pandemic also poses psychological challenges. For example, (self) quarantine and isolation may seem similar; but (self) quarantine is separation for people who were actually or plausibly exposed to a contagious disease (such as from international travel) but are not confirmed to be infected, whereas isolation is for people who are infected with a contagious disease [13]. As such, (self) quarantine may seem less stressful, not only because the person is presumably not infected, but also because the person is supposedly in control of their quarantine. Yet the stress of not being tested (as in many jurisdictions only symptomatic people or people who have been in contact with infected people are tested) may worsen the (self) quarantined person's stress. Also, the social pressure—and the legal requirement in an increasing number of jurisdictions—to (self) quarantine may reduce the person's sense of control and even generate distress related to the discrepancy between social expectations and individual entitlement to freedom of movement (in jurisdictions where that is legally supported). Hence, the highly prevalent psychological needs for certainty and for sense of control are not easily addressed in self-quarantine and may require particular attention during the pandemic to facilitate mental well-being of (self) quarantined people. Another example is the likely loss of trust in people who are physically close (and personally significant) to a person in case they are either infectious (while asymptomatic) or are not careful enough in trying to prevent being infected. Such a pervasive loss of trust may deeply disrupt people's mental well-being and functioning, particularly if they are already vulnerable such as having an insecure attachment style [14]. Hence, the universal psychological need for trust is not easily addressed with family and friends during the pandemic, particularly in relation to emotionally vulnerable people, and may require particular attention during the pandemic to facilitate their mental well-being and functioning. These and other psychological challenges related to the pandemic period may last beyond it, especially if there were personally traumatic events during it, such as forced self-quarantine by authorities and betrayal of trust by (personally) significant others. These challenges may require special attention after the pandemic to facilitate mental well-being and functioning of people who are identified as having—or being at high risk of having—pandemic-related mental problems after the pandemic.

## 3. Philosophical (Ethical and Epistemological) Reflections

Some of the pandemic-related social and psychological issues are associated with underlying ethical issues. An example is the scarcity of health care resources, which is rampant during the current pandemic, as it has been during some other pandemics, such as the Spanish flu pandemic (when human health resources—particularly physicians and nurses—in the United States were depleted due to their deployment abroad near the end of World War I [15]); decisions about which treatable patients to exclude from treatment—such as ventilation—can cause moral distress and other disruption to health care providers. Hence health care providers may require particular attention during and after the pandemic to address their moral distress. Another example is the common—personal and social—expectation during the pandemic that individuals help others, above and beyond what is expected in more ordinary times. Although ordinary ethics would consider that as supererogatory, i.e., laudable but not required morally, during extraordinary—such as pandemic—times, extraordinary moral conduct may

be expected if not required, involving increased individual moral responsibility [16]—including for others' plight even if there is no preset relationship between them and the individual expected to help them. Hence, the general public may require particular attention during and after the pandemic for emotional and practical support in relation to such extraordinary moral conduct expectations.

Some of the pandemic-related social and psychological issues are also associated with underlying epistemological issues. For example, individual and collective behavior impact biological aspects of the pandemic such as rate of transmission, yet robust evidence on that is difficult to obtain due to the lack of availability of randomized controlled trials in such circumstances. Other approaches to generate robust evidence are needed in these circumstances, such as studies comparing naturally variant sites and populations and sufficiently matched samples, recognizing that comparison is key to any inquiry [5]. Hence, researchers may require particular attention during the current pandemic and in preparation for future pandemics and other global health crises to optimize their research methodology for such circumstances. Another example is the common misunderstanding by lay people of what is robust evidence, which may lead to their unsafe behavior or alternatively to their overly cautious behavior during the pandemic. This may pose unnecessary personal harm and public risk (due to increased transmission of infection) or alternatively unnecessary personal restriction and social disruption (due to unnecessary reduction of work and other activities), respectively. Hence, the general public may require particular attention during the current pandemic (and arguably at all other times) to enhance lay people's critical thinking and knowledge about evidence and other relevant aspects of rigorous inquiry such as health research.

## 4. Conclusions

Social, psychological, and underlying philosophical issues that are pandemic-related may have a considerable and lasting impact on societies and on particular individuals. Some related practice suggestions are to address the moral distress of health care providers who have to make particularly difficult—sometimes life or death—decisions due to very scarce health care resources, and to provide additional emotional support such as to (self) quarantined people and to people who have pre-pandemic mental challenges (preferably provided by their significant others and/or mental health care providers). Some related policy suggestions are to secure additional income support for socially disadvantaged people during and soon after the pandemic, and to provide additional protections for special social groupings that are considered socially undesirable by some if the pandemic results in disruptive political change (that may last after the pandemic). Some related research suggestions are to study societal preparation for pandemics, perhaps learning from positive deviance such as Taiwan's successful preparation for the current (COVID-19) pandemic based on its experience with the SARS pandemic nearly 20 years ago [1]. Some related education suggestions are to train the general public as well as health care providers and other first responders in advance in responsible behaviors that protect them and others during a pandemic and other challenging times. More reflection on and study of social and psychological challenges as well as underlying philosophical issues related to the current pandemic, and more generally to global crises, is needed.

**Funding:** This research received no external funding.

**Conflicts of Interest:** The author declares no conflict of interest.

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
