# Peer review of "Social, Psychological, and Philosophical Reflections on Pandemics and Beyond"

_societies, doi:10.3390/soc10020042_

Round 1

Reviewer 1 Report

This is a very interesting manuscript and the topic is very important. 

However, it is underdeveloped and there are some issues that need to be addressed before it can be considered for publication. 

  1. The abstract is very short and vague. E.g. “Suggestions are made in relation to current and future relevant practice, policy, research, and education”

Please elaborate about the suggestions made.

Please also elaborate on what types of policy, practice, and research you refer to?

Surely the paper does not reflect every policy context that exists, and all research agendas?

  • Generally, abstracts are 150 to 200 words, often with an upper limit of 250 words. Please refer to the journal instructions and ensure that the abstract meets the specific guidelines.

  • The introduction is too broad and vague. For example, the first sentence states “Societies are measured in part in relation to how they rise to the occasion of collective crises.” How have societies risen to collective crises? What do you mean? Since you are discussing Covid19, examples that might be discussed are SARS

  • Lines 15 to 16 discusses “many problems that are insufficiently addressed to date”. Please elaborate on what types of problems you mean.

  • Lines 24-25 repeats the same broad statement about “practice and policy as well as research and education suggestions”. Please be more specific

  • Please number the pages.

  • Lines 29-30 refers to “insufficient work load, such as in the retail sector.” Please cite any sources for which you came to this conclusion. From my understanding, there are many parts of the retail sector that are flourishing at the moment, such as online sales through Amazon, as well as grocery retail. There also seems to be a spike in demand for delivery men and women to purchase goods and deliver them given to people who want to avoid in-person shopping.

  • Lines 35 and 44 refer to “socially disadvantaged people” and “socially undesirable” people.  Who are these people? Are they of certain gender, race, and/or class? If so, it might be a good idea to state add on “such as women, racialized people, etc.” explicitly.

  • Line 38 refers to “countries where the regime is not democratically robust”. Which country are you referring to? Citations would be helpful.

  •  Lines 41 and 44 repeats the words “by some”.       Consider deleting it in line 41, as it is unnecessary.

  • Lines 48 and 49 refer to “self-quarantine” and “ isolation” as unique definitions. Please cite the sources used.

  • Line 54 starts the sentence with “And”. For an academic, scholarly paper, it is too informal to start a sentence with a conjunction. See https://blog.esllibrary.com/2017/08/17/can-we-start-a-sentence-with-a-conjunction/

Please revise the statement.

  • Line 61 mentions loss of trust in in people who are “close to a person”. Please elaborate on what you mean by this. Later in the same paragraph, you refer to “significant others” (Line 71). Is this what you mean? If that is the case, please revise accordingly as it is not clearly stated in the beginning of the paragraph.

  • Line 77-78 refers to “scarcity of health care resources” that are “rampant during the current pandemic, as has been during other pandemics”. Please elaborate which pandemics you are referring to? Spanish Influenza of 1918? Bubonic Plague? Pneumonic Plague? I would caution such a sweeping statement, as these earlier pandemics were not necessarily about scarcity of health care resources but rather under-developed health and social care.

  • Line 93 refers to “such as rate of contagion”.

Please revise as this seems grammatically incorrect. A contagion has a rate of transmission. Is that what you mean, rate of transmission?

  • Lines 102 to 104 refers to “…their overly cautious behavior during the pandemic, posing unnecessary personal harm and public risk…”

What do you mean by this? How does overly cautious behavior pose public risk? Risk of what? Risk of contracting Covid19? If the latter is the case, then I would suggest that a citation is required, because currently, the evidence indicates that being overly cautious is flattening the curve, and is actually a public benefit and public good, and not at all a public risk.

  • Line 117 has a grammatical error with respect to subject/verb agreement. “Some policy suggestion are …” it should be “Some policy suggestions are…”

  1. Line 122 mentions Taiwan’s case. Please cite the source of this information.

  • Line 126 also begins with a conjunction, “And”.       As indicated in comment 12 above, for an academic, scholarly paper, it is too informal to start a sentence with a conjunction. See https://blog.esllibrary.com/2017/08/17/can-we-start-a-sentence-with-a-conjunction/

Please revise the statement.

  • Please revise the manuscript with tracked changes and upload this version so that the reviewer may see the changes clearly.

Author Response

Thank you to this reviewer for his/her comments and suggestions. Below find my responses to all his/her comments and suggestions (recognizing that line numbers have changed with my revision based on these comments and suggestions):

  1. In response to the reviewer's suggestion to lengthen the Abstract and as part of that elaborate on my suggestions, such as in relation to policy and practice, lengthened the Abstract to more than 100 words and as part of that I have added that such as in relation to Taiwan's learning from SARS in comparison to Canada's learning from SARS. 
  2. In response to the reviewer's suggestion to add more specifics to the Introduction, I have done so, such as in relation to Taiwan's learning from SARS in comparison to Canada's learning from SARS. 
  3. In response to the reviewer's suggestion to elaborate on problems in lines 15-16, I have illustrated that in relation to the impact of international travel on global health. 
  4. In response to the reviewer's suggestion to add some specifics related to practice and policy as well as research and education suggestions in lines 24-25, I have deferred that to later after the Introduction. 
  5. In response to the reviewer's suggestion to add page numbers, I defer that to the publisher as pagination may change in press. 
  6. In response to the reviewer's suggestion to clarify how the retail sector is impacted, I have changed that to the service sector which is a more obvious example.
  7. In response to the reviewer's suggestion to specify social disadvantage and social undesirability in lines 35 and 44, I added examples of homeless people and racialized minorities. 
  8. In response to the reviewer's suggestion to illustrate a country that is not robustly democratic in line 38, I added the example of Israel.
  9. In response to the reviewer's suggestion to delete the repetitive word "some" in lines 41 and 44, I changed wording of that sentence to do that. 
  10. In response to the reviewer's suggestion to reference definitions of (self) quarantine and isolation in lines 48 and 49, I added such a reference. 
  11. In response to the reviewer's suggestion to reword the sentence in line 54 so that it does not start with the word And, I revised the sentence in such as way that it does not start with the word And. 
  12. In response to the reviewer's suggestion to reword the term "close to a person" in line 61 so that it is more clearly related to the term "significant others" that appears later, I reworded that sentence to address personal significant and physical closeness. 
  13. In response to the reviewer's suggestion to address in lines 77-78 whether other pandemics such as the Spanish flu had scarce health care resources or underdeveloped health care resources, I added clarification addressing scarce (human) health care resources in the United States during the Spanish flu. 
  14. In response to the reviewer's suggestion to replace the word contagion with the word transmission in line 93, I have done that in the revision. 
  15. In response to the reviewer's suggestion to clarify what is overly cautious behavior and related public risk in lines 102-104, I added the example of limiting work and other activities. 
  16. In response to the reviewer's suggestion replace the singular word suggestion [which was a typo) with the plural word suggestions in line 117, I replaced that in the revision. 
  17. In response to the reviewer's suggestion to reference the information about Taiwan in line 122, I added that reference in the revision. 
  18. In response to the reviewer's suggestion to reword the sentence in line 126 so that it does not start with the word And, I did that in the revision.  

Reviewer 2 Report

Social, Psychological and Philosophical Reflections on Pandemics and Beyond

The manuscript discusses certain well-known examples from the the current pandemic news, conditionally classified into three sections - social, psychological and philosophical issues.

Please consider the following comments.

line 8
Please provide more specific wording of the proposals in the abstract.

l.42
The sentences contain repetitions of phrases.
(...) and after it (…) after the pandemic.

l.68, l.119
The text contains excessively long sentences (52 words, 72 words each), in which there are repetitions of words and phrases. Please, consider revision.

In accordance with the requirements of Societies journal for manuscripts, the article types are as follows: articles, short communications, reviews, conceptual papers. The manuscript indicated by the author as “essay” potentially refers to “conceptual papers”. However, I found neither "new ways of looking at existing knowledge and concepts" nor "new research questions."

Seven out of ten references (No. 2-6, No. 9-10) are the author's publications. The rate of self-citation is 70%. Please, consider revision.

Author Response

I thank this reviewer for his/her comments and suggestions. In full response (recognizing that line numbers have changed in the revision; I address the original line numbers below):

  1. I provided more specific detail in the proposals in the Abstract in line 8, such as in relation to addressing socially disadvantaged populations. 
  2. I deleted repetition of phrases ...and after it... after the pandemic... in line 42. 
  3. I shortened sentences, e.g., in lines 68 and 119.
  4. I clarified in the Abstract and in the Introduction that this conceptual paper is analytic and comparative (more than synthetic/innovative, although analysis and comparison lead to conclusions that may be innovative).
  5. I added more references that are not my publications, so that in the revision less than half of the references are my publications.   

Round 2

Reviewer 1 Report

This is a much improved version of the manuscript. 

One final suggestion would be to revise the abstract further in line 9: Line 9 also states "addressing Taiwan's better learning from SARS". This is not clear. Better than what? Is Taiwan's model advocated in this paper? I.e. does it mean the paper accomplishes to "advocate Taiwan's experience from SARS as a learning experience for Canada?"

Line 121 to 126 is a run-on sentence that is nearly 5 lines long. It might be better to break it down into two or more sentences rather than inserting parantheses ().  

Line 88 requires a period between "others" and "The"

Author Response

Thank you to this reviewer for further comments. Below I address them:

  1. In line 9, I have now deleted the word better to not imply that Taiwan has learned better than Canada from SARS; in the text I elaborated on that in a more nuances manner (such as in relation to positive deviance) in the previous version - that is still in place in the text.  
  2. In lines 121-126, as suggested, I have now broken down the long sentence to two sentences. 
  3. In line 88, as suggested, I have now added a period between the word others and the word Then. 

Reviewer 2 Report

I still believe that the author inappropriately uses self-citations (more than 40% of self-citations), but this is at the discretion of the editors of the Societies journal.

Author Response

I thank this reviewer for the further comment about my self-citations. In this case I beg to differ, as for such a broad scope paper, authorship credibility related to inter-disciplinarity and multiple expertise is helpful for readership, considering that many single authors would not have such credibility.